# A method for structure prediction of metal-ligand interfaces of hybrid nanoparticles

Sami Malola[1], Paavo Nieminen [2], Antti Pihlajamäki[1], Joonas Hämäläinen[2], Tommi Kärkkäinen[2] & Hannu Häkkinen [1,3]

Hybrid metal nanoparticles, consisting of a nano-crystalline metal core and a protecting shell of organic ligand molecules, have applications in diverse areas such as biolabeling, catalysis, nanomedicine, and solar energy. Despite a rapidly growing database of experimentally determined atom-precise nanoparticle structures and their properties, there has been no successful, systematic way to predict the atomistic structure of the metal-ligand interface. Here, we devise and validate a general method to predict the structure of the metal-ligand interface of ligand-stabilized gold and silver nanoparticles, based on information about local chemical environments of atoms in experimental data. In addition to predicting realistic interface structures, our method is useful for investigations on the steric effects at the metal-ligand interface, as well as for predicting isomers and intermediate structures induced by thermal dynamics or interactions with the environment. Our method is applicable to other hybrid nanomaterials once a suitable set of reference structures is available.

[1] Department of Physics, Nanoscience Center, University of Jyväskylä, FI-40014 Jyväskylä, Finland. [2] Faculty of Information Technology, University of Jyväskylä, FI-40014 Jyväskylä, Finland. [3] Department of Chemistry, Nanoscience Center, University of Jyväskylä, FI-40014 Jyväskylä, Finland. Correspondence and requests for materials should be addressed to T.K. (email: tommi.karkkainen@jyu.fi) or to H.H. (email: hannu.j.hakkinen@jyu.fi)

Hybrid metal nanoparticles, consisting of a nano-crystalline metal core and a protecting layer (shell) of organic ligand molecules, are an emerging class of functional nanomaterials that have potential applications in diverse areas such as biolabeling, catalysis, nanomedicine, and solar energy[1–8]. The core-shell framework structure of hybrid nanoparticles offers ample opportunities to tune the physico-chemical properties and functionalities of the particles via controlling the size, shape, elemental composition, and structure of the metal core, together with the chemical composition of the ligand shell. The chemical interactions between the metal atoms and ligand molecules at the core-shell interface are in a crucial role since they dictate the atomic-scale structure, stability, and ensuing properties of the particle.

The last decade has witnessed clear advancements in synthesis and experimental structural characterization of very small, atomically precise hybrid nanoparticles with 1–3 nm cores made of metals, stabilized by various organic ligands[3]. These particles are also called monolayer-protected clusters, MPCs, and they represent an interesting subclass of nanoparticles since their structures can be often characterized to atomic precision by using X-ray diffraction method on single MPC crystals. At the moment, more than 150 crystallographically solved structures of MPCs, involving noble metals, main group metals, and various ligand molecules such as thiols, phosphines, and alkynyls, have been reported[3]. This facilitates fundamental studies of the structure-property relationships both experimentally and computationally.

In most cases, however, the knowledge of the nanoparticle structures does not reach the atom-level resolution and the ligand-metal interfaces may be ill-defined. Only partial structural knowledge may be available, e.g., by high-resolution electron microscopy where only the heavy atoms (metals) of the core may be visible[9,10]. Smallest particles may have low-symmetry or disordered metal cores, and may not be amenable at all to experimental techniques that work well for structural characterization of atomically ordered bulk materials[11]. A practical solution is then to reach conclusions of potential atomic-scale structures by comparing measured properties, such as powder X-ray diffraction data, and various spectroscopic data, to computed properties based on extensive sets of potential candidate structures. A crucial question is then how realistic is the group of the candidate structures, i.e., can the structure corresponding to the true global total energy minimum be included in that group with a high probability. In general, global optimization methods suffer from limitations arising from a prohibitively (exponentially) increasing number of local energy minima in the structural space for system sizes that are larger than just a few metal atoms and ligand molecules. Another time-constraint arises from the fact that most measurable properties must be calculated numerically from the electronic structure using the platform of the density functional theory (DFT), which limits the number of structural candidates that can be examined. It is thus crucial to develop methods that can effectively suggest realistic atomic-scale structures at a very low computational cost.

The data on atomically precise structures of MPCs, combined with an ever-growing number measurements of their physico-chemical properties, collectively contains valuable chemical information on the atomic bonding and structure-property relations of these nanomaterials, which could be used for successful structural predictions of yet unknown nanoparticles. Here, we devise and demonstrate a general method for predicting metal-ligand interface structures of an unknown ligand-protected metal nanocluster. Our method is based on a local search algorithm that uses information about the known local atomic environments at the metal-ligand interface of reference nanostructures in the same class of hybrid nanoparticles. The specific example systems discussed in this work comprise gold (Au) and silver (Ag) nanoclusters protected by thiols (SR), phosphines (PR₃) and diphosphines (DPPY), and we demonstrate how experimentally verified Au/Ag-thiolate and Ag-phosphine interface structures can be successfully built when the positions of all Au or Ag atoms in the cluster are first defined. However, the method itself is general and can be used for any type of nanoparticle or nanostructure if enough reference structural information is available in the same class of systems. The set of reference structures can be considered as a training set and the whole procedure to refine the candidates for the metal-ligand interfacial structures may be considered as an analogue to applying machine learning methodology to the structure prediction problem[12].

## Results

**Structure prediction algorithm.** Our procedure to build candidates for the atomic structure of metal-ligand interfaces is summarized below and illustrated in Fig. 1. Supplementary Note 1 describes the algorithms 1–4 involved. The main algorithm 1 is divided into steps 0–4 as follows (the steps are also numbered the same way in Fig. 1).

0. First, a group of known reference structures (training set) is defined. The training set may include experimental crystal structures, computational model structures, partial structures, or hand-made intuitive structural guesses.

1. The coordinates for the metal atoms (here Au or Ag) are set. This information may come, e.g., from experimental electron microscopy data.

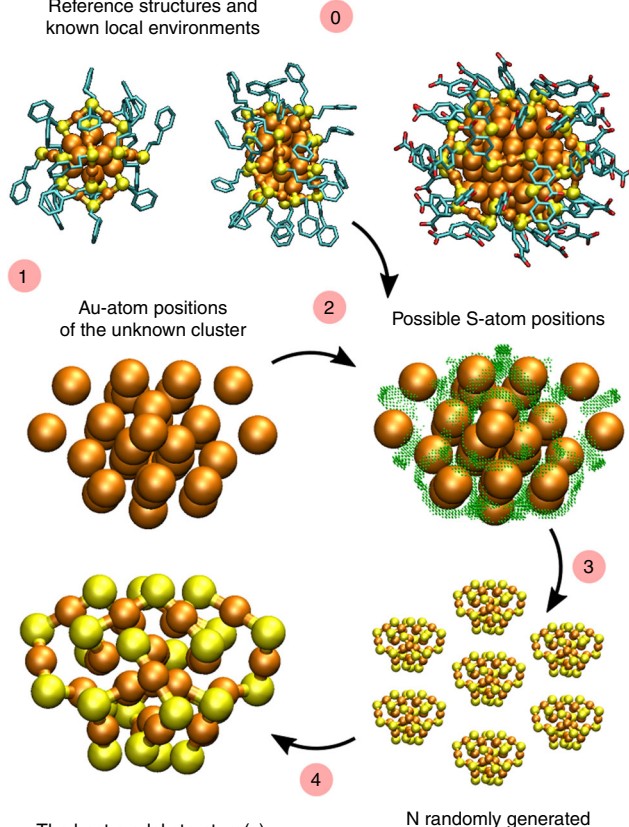

**Fig. 1** A schematic visualization of the algorithm. Using a set of reference structures of known $Au_x(SR)_y$ nanoclusters, several candidates for the structure of the layer of sulfur atoms are built around the gold core of the unknown cluster, with selective ranking of the most probable structures. Au: orange, S: yellow, carbon backbone in ligands: cyan

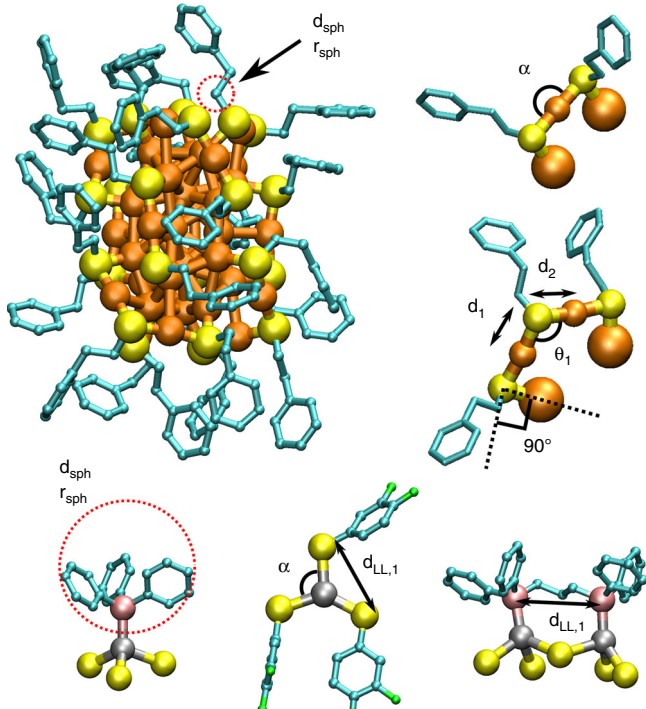

**Fig. 2** Parameters related to the local environment of S-atoms. $d_1$ and $d_2$ are the two nearest neighbour Au-S distances and $\theta_1$ is the selected bond angle between S-atom and the two nearest neighbour Au-atoms. Criteria for the S-Au-S and S-Ag-S bond angles ($\alpha$) close to 180 degrees are used for adding or recognizing the atoms in a linear SR-Au-SR and SR-Ag-SR conformations and 120 degrees for three-coordinated Ag-S complexes. Spatial fitting of ligands is tested with a rigid sphere (red dashed circles) in a perpendicular direction to the two nearest neighbor Au-S bonds of each S-atom and parallel to Au-P bond for phosphines. Distance of the sphere from the binding site S-atom is also defined. Here, the sphere mimics the first $CH_2$ group of phenyl ethane thiol $SCH_2CH_2Ph$ and the whole $Ph_3$ group for $PPh_3$ ligand. $d_{LL,1}$ represents the distances between of the added interface atoms which are restricted by parameters when adding special local structures or when limiting the smallest possible interface atom distances. Colors: Ag: gray, F: green, P: pink and the rest as in Fig. 1

2. A list of possible positions for interface atoms (here S or P) around the metal core is built. The local chemical bonding determines the acceptance of a point of a dense grid (Supplementary Fig. 1) as a possible interface atom position, based on comparison to bond lengths and angles found in the training set (Supplementary Fig. 2). In addition, a spatial fitting of the molecule (ligand) attached to the interface is required for acceptance. Here, fitting of the organic part of thiolates and phosphines is tested with a specific sized rigid sphere in the most prominent bonding direction with respect to the nearest neighbor Au–S, Ag–S or Ag–P bonds as visualized in Fig. 2. Finally, the list of all possible interface atom positions is saved and used as a basis for the random multi-start selection process in step 3.

3. N complete interface structures are created from the possible interface atom positions by a restricted random multi-start selection process as described in Algorithms 2–4 (Supplementary Note 1). The restrictions are based on simple rules of chemistry related to coordination, atomic distances and local conformations (Supplementary Note 2 and Fig. 2). The process starts by adding randomly picked interface atoms, from the list generated in step 2, first into specified local conformations such as linear SR–Au/ Ag–SR arrangements resembling a part of protecting SR-(Au/Ag-SR)$_n$ ($n = 1, 2$) units (Fig. 2). The protecting units are the main

components forming the gold-thiolate interface structure in many known thiolate-protected gold clusters. For protected Ag-clusters, more than one local structural arrangement is possible with respect to SR–Ag–SR angles because of the more flexible Ag–S coordination compared to gold (see Fig. 2 and Supplementary Fig. 3). In the case diphosphines, the specified local conformation is a pair of phosphorus atoms, for which the P–P distance is restricted based on the connecting organic group between the phosphine ends of the molecule. Next, single thiolates or single phosphines will be used to fill the remaining free metal surface if necessary. The selection process ends when there are no valid points left in the list of possible interface atom positions. All selected interface atom coordinates form one potential model structure. The total number of generated model structures should be set large enough to overcome the challenges of a non-guided stochastic process to represent all relevant overall interface conformations.

4. The final step of the algorithm ranks the generated model structures. For the ranking we introduce a numerical criterion, a so-called combined structural error (CSE) (see the full description in Supplementary Note 3), which is constructed as an average from the errors related to the nearest neighbor bonds and angles, to the number of predicted interface atoms in specified local conformations, and to the total number of interface atoms. Experimental evidence indicates that the most stable ligand-protected gold and silver clusters also have the most complete steric protection of the metal core by the ligand layers. After ranking, the best model structures can be completed by adding the organic part of the ligand layer and optimizing the atomic structure of the ligand layer by molecular mechanics or molecular dynamics. One possible approach is demonstrated in our previous work[13].

A working strategy to avoid stochastic challenges (non-effective random search of possible ligand atom positions for larger clusters) is discussed in Supplementary Note 4. Criteria for selecting the geometric parameters for step 2 above are discussed in Supplementary Note 2.

**Reference structures and validation of the algorithm.** For Au–S interface prediction, the set of reference structures included 24 known protected gold nanoclusters Au$_x$(SR)$_y$ between the sizes $(18,14) \leq (x,y) \leq (279,84)$ as well as the short RS–Au–SR and longer RS-Au-SR-Au-SR unit conformations. For Ag–S and Ag–P interfaces, 17 known thiolate and/or phosphine-protected silver nanoclusters were included in the set of reference structures. The reference silver nanoclusters have 14–374 Ag-atoms and 20–117 ligands including thiolates, phosphines, diphosphines and halides. A complete list of the reference structures is given in Supplementary Tables 1 and 2. Based on the reference information, the algorithm was validated for protected Au–S interfaces by examining 10 Au$_x$(SR)$_y$ clusters in the range $(34,22) \leq (x,y) \leq (279,84)$ where the structure is known experimentally[14–23], removing the ligand layer including sulfurs, by building a large number of potential structures of the ligand layer around the fixed gold core, ranking the structures, and comparing these Au–S interface structures with the experimental crystal structure. Validation in the case of Ag–S interfaces was done with four protected Ag-clusters in the size range of 23–211 Ag-atoms and 26–78 ligands[24–27]. Two of these clusters had ligand layers consisting of both thiolates and phosphines, and one consisting of thiolates and diphosphines. Combined, the selected Au- and Ag-clusters include various different symmetries, cluster sizes, surface curvatures, surface morphologies, and protecting ligand motifs. Table 1 and Supplementary Tables 3–8 give the full structural details and parameters related to the prediction and validation.

**Table 1 The results of predicting sulfur positions for ligand-protected Au- and Ag-clusters**

| Cluster | $N_{models}$[a] | $N_{max,unit}$[b] | $N_{max,tot}$[c] | $N_{correct}$ %[d] | Min $_{RMSD}$ [Å][e] |
|---|---|---|---|---|---|
| $Au_{34}(SR)_{22}$ | 9216 | 22 | 22 | 28.0 % (2579) | 0.261 |
| $Au_{36}(SR)_{24}$ | 9216 | 24 | 24 | 6.66% (614) | 0.432 |
| $Au_{38}(SR)_{24}$ | 9216 | 24 | 24 | 94.0% (8665) | 0.383 |
| $Au_{44}(SR)_{26}$ | 9216 | 24 | 25–26 | 85.2% (7853) | 0.339 |
| $Au_{52}(SR)_{32}$ | 9216 | 32 | 32 | 9.39 % (865) | 0.319 |
| $Au_{92}(SR)_{44}$ | 9216 | 36 | 39–44 | 1.79% (165) | 0.525 |
| $Au_{102}(SR)_{44}$ | 9216 | 44 | 44 | 5.41% (499) | 0.395 |
| $Au_{146}(SR)_{57}$ | 9216[f] | 50 | 54–60 | 1.21 % (121) | 0.545 |
| $Au_{279}(SR)_{84}$ | 9216[f] | 60 | 75–84 | 0.67 % (62) | 0.539 |
| $Ag_{23}(SR)_{18}(PPh_3)_8$ | 9216 | 18 | 18 | 1.19 % (110) | 0.225 |
| $Ag_{44}(SR)_{30}$ | 9216 | 30 | 30 | 42.9 % (3958) | 0.375 |
| $Ag_{78}(SR)_{42}(DPPP)_6$ | 9216 | 40 | 41–42 | 0.022 % (2) | 0.644 |
| $Ag_{211}Cl(SR)_{71}(PPh_3)_6$ | 9216[f] | 70 | 70–72 | 0.85 % (78) | 0.744 |

[a]Number of model structures
[b]Maximum number of sulfurs in units
[c]In total if maximum number of sulfurs in units
[d]Percentage of correct structures. In the case of Au clusters, the criterion to have the correct structure is that 2/2 of nearest neighbors are correct for each atom. In the case of Ag clusters, the criterion is 4/5 of nearest neighbours to be correct for each atom
[e]Min RMSD of the correct structures
[f]Iterative prediction runs

The cluster that was predicted was excluded from the reference structures in each example.

**Au–S interface structure for gold clusters**. To test and validate our algorithm for Au–S interfaces, we selected nine known clusters: $Au_{34}(SR)_{22}$ (ref. [14]), $Au_{36}(SR)_{24}$ (ref. [15]), $Au_{38}(SR)_{24}$ (ref. [16]), $Au_{44}(SR)_{26}$ (ref. [17]), $Au_{52}(SR)_{32}$ (ref. [18]), $Au_{92}(SR)_{44}$ (ref. [19]), $Au_{102}(SR)_{44}$ (ref. [20]), $Au_{146}(SR)_{57}$ (ref. [21]), and $Au_{279}(SR)_{84}$ (ref. [22]). In addition, $Au_{44}(SR)_{28}$ cluster[23] was also used for analyzing the ligand size effects at the interface as discussed later. The summary of the parameters used in prediction are given in Supplementary Table 3. The test systems include various different ligand layer conformations consisting of different length of SR-(Au-SR)$_n$ protecting units and single bridged thiolates. For all clusters, 9216 model structures were generated in a single run, and for the two largest clusters $Au_{146}(SR)_{57}$ and $Au_{279}(SR)_{84}$ the prediction was done twice by combining the sulfur atom positions of 20 best model structures of the first round into a new set of possible S-atom positions for the second round. Twenty best model structures were taken from those that had the largest number of ligands in units and in total in the first round. Furthermore, correct structures were determined by comparing the atomic indices of the two nearest neighbor Au-atoms of each sulfur of the model structure with respect to the nearest sulfur atom of the true known structure.

Table 1 shows the results. The success of prediction was determined as a ratio of the correct structures to all generated model structures, varying notably much, from 0.67 to 94.0%. The maximum number of S-atoms found in the protecting units and in total are in a very good agreement with the compositions of the true structures. This indicates that the true stable structures maximize the number of ligands on the surface to best protect the metal core from the degradation. The lowest RMSD (root mean square deviation) values of the correct model structures range from 0.261 to 0.545 Å, and the corresponding structures are shown in Fig. 3. The algorithm is flexible to be used with different kind of sets of reference structures, as indicated for $Au_{102}(SR)_{44}$, where only conformations of one short SR-Au-SR unit and one long unit SR–Au–SR–Au–SR unit were used successfully for the prediction. For all other systems the complete set of reference structures were used omitting always the cluster in question.

The ratio of correct S-atom positions as a function of CSE is shown for all of the examined protected Au-clusters in Fig. 4. The

correctly positioned atoms were determined similarly as in Table 1. For all systems in panels a–i of the Fig. 4 the ratio of correct atoms approaches 1.0 when CSE gets smaller. For most of the investigated systems, the model structure with a minimum CSE value matches with the true structure by nearest neighbor bonding, except in the most challenging case of $Au_{279}(SR)_{84}$, where the true structure is found among 5–10 best model structures. For the CSE, five nearest neighbors were used for describing the local environment of atoms and for all systems the corresponding error was calculated with respect to the same set of reference structures that was used in the model structure generation. Noteworthy is that the CSE separates the correct structure of $Au_{102}(SR)_{44}$ regardless of the fact that the set of reference structures is considerably smaller, although the observed range of the CSE, 0.26–0.46, reflects the incompleteness of the set of reference structures used in the prediction. For the clusters which have comparable decent set of reference structures, the minimum CSE values are found consistently between 0.04 and 0.06 regardless of the size of the system. The ratio of correct atoms on the surface depends on the other hand on the complexity of the investigated system. For the simplest systems such as $Au_{38}(SR)_{24}$, all the generated model structures have >85% of the added S-atoms correct but for example for the similar sized cluster $Au_{36}(SR)_{24}$ the range is 30–100%.

The structures of $Au_{44}(SR)_{26}$, $Au_{92}(SR)_{44}$, $Au_{146}(SR)_{57}$, and $Au_{279}(SR)_{84}$ have single bridged ligands that do not resemble the arrangement of protecting units. The number of bridged ligands added on the surface of each of these clusters is in the range of 2–18 and can be seen from the difference of the number of ligands in total and in the units shown in Table 1. For some of the systems the total number of the interface atoms added on the surface exceeds the true number of ligands, but regardless of that the algorithm is accurate enough for predicting both the number and the positions of the bridge ligands by the ranking criteria. These results confirm that in the true structures the number of interface atoms in linear SR–Au–SR conformations is often maximized. This is automatically taken into account in the design of the algorithm and is also build into the CSE measuring the goodness of the model structures.

**Ag–S interface structure for silver clusters**. The main difference when predicting Ag–S interface as compared to predicting Au-S interface is to allow more flexible coordination of the metal atoms

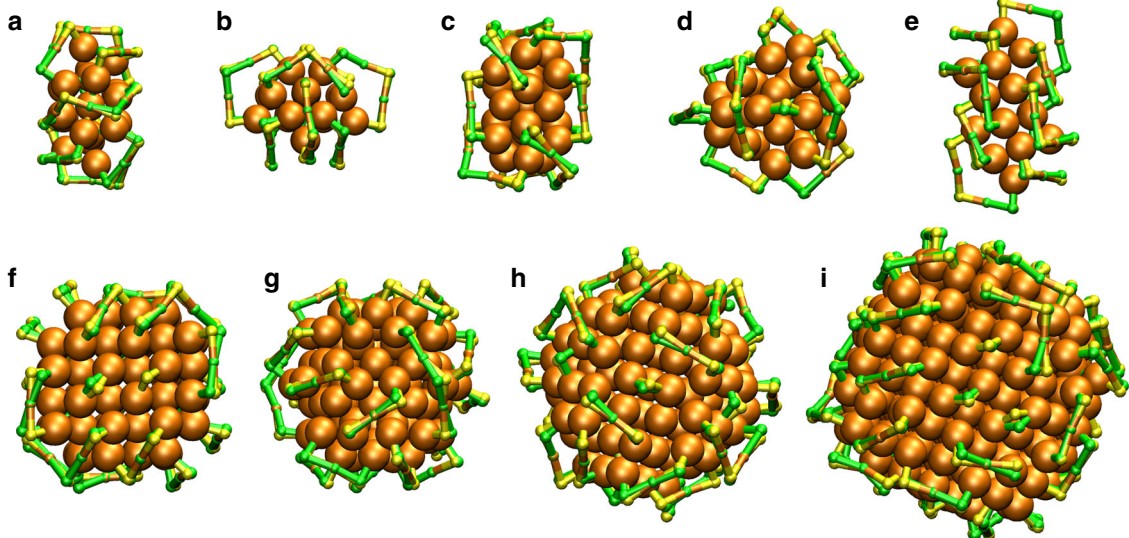

**Fig. 3** Validation of structure prediction for Au-clusters. Comparison of the predicted and true ligand-metal interface structures of protected Au-clusters: **a** Au$_{34}$(SR)$_{22}$ (ref. [14]), **b** Au$_{36}$(SR)$_{24}$ (ref. [15]), **c** Au$_{38}$(SR)$_{24}$ (ref. [16]), **d** Au$_{44}$(SR)$_{26}$ (ref. [17]), **e** Au$_{52}$(SR)$_{32}$ (ref. [18]), **f** Au$_{92}$(SR)$_{44}$ (ref. [19]), **g** Au$_{102}$(SR)$_{44}$ (ref. [20]), **h** Au$_{146}$(SR)$_{57}$ (ref. [21]), and **i** Au$_{279}$(SR)$_{84}$ (ref. [22]). Predicted S-atoms and Au–S bonds are drawn in green and true in yellow. Au-atoms are drawn with orange color

to thiolates. The selected parameters for protected Ag-clusters can be seen in Supplementary Table 4. For validating the prediction of Ag–S interface structures we selected four known protected Ag-clusters: Ag$_{23}$(SR)$_{18}$(PPh$_3$)$_8$ (ref. [24]), Ag$_{44}$(SR)$_{30}$ (ref. [25]), Ag$_{78}$(SR)$_{42}$(DPPP)$_6$ (ref. [26]), and Ag$_{211}$Cl(SR)$_{71}$(PPh$_3$)$_6$ (ref. [27]). For these clusters most of the ligands on the surface are thiolates, but also phosphines, diphosphines and one halide are included. The single halide is omitted in the prediction. Our approach to predict the interfaces with mixed ligands is to predict first the positions of the ligands that are in majority and then to continue by predicting the other ligands, in this case phosphines and diphosphines. For all test systems, 9216 model clusters were generated during a single run. The prediction was done twice for the largest Ag$_{211}$Cl(SR)$_{71}$(PPh3)$_6$ cluster by combining the S-atom positions of 20 best structures of the first run to the set of possible S-atom positions for the second run, similarly to the two largest Au-clusters discussed above. The correctly positioned atoms were determined by requiring that four out of five nearest neighbors metal atom indices must be correct for all interface atoms as compared to the true structure. This criterion is different than in Au–S interfaces due to the enriched bonding configurations on Ag–S interface, for which taking into account only two nearest neighbors would not be enough.

Table 1 shows the results from prediction for all four systems. The success of the prediction varies from 0.022 to 42.9%. To remark, the success ratio increased from 0 to 0.85% for the largest cluster Ag$_{211}$ from the first to the second prediction run. The maximum number of ligands found in protecting units and in total are in a very good agreement with the molecular compositions of the true structures for all systems. The minimum RMSD values of the correct structures are in a range of 0.225–0.744 Å and the corresponding structures are shown in Fig. 5. The Ag–S interfaces are more complex than Au–S interfaces especially in larger clusters which can be seen as a more significant variations in the positions of the sulfur atoms compared to the true structure and also in the success rates and the RMSD values.

CSE is efficient also for predicting the true structures for protected Ag-clusters as can be seen from Fig. 6 for the four studied cases. Similarly to Au-clusters, the model structures that

have the smallest CSE values include most probably the true structure. It is interesting to note that even for the clusters with mixed ligand layers of thiolates and phosphines the true Ag–S part of the overall interface conformation can be predicted before addition of the phosphines or diphosphines. This enables prediction of the overall conformations of the metal-ligand interfaces in steps, first for the ligands that are in majority and then continuing the process with the ligands in minority for the best structures. Completing the prediction with phosphines and diphosphines is described next.

**Phosphine positions for the mixed ligand silver clusters**. Predicting Ag–P interface for the clusters with mixed ligand layers of thiolates and phospines or thiolates and diphosphines was done by starting from one of the correctly predicted partial structures based on the Ag-S interface predictions. Both Ag-atom and S-atom positions are included when describing the local environments for the phosphorus atoms. To accomplish this, atom types were also included the nearest neighbor description when searching the possible interface atom positions. The parameters used in prediction of Ag–P interface are given in Supplementary Table 5. For all clusters we used systematically four nearest neighbors with 0.1–0.2 Å error limits for the distances and 10 degrees limit for the angles. One of the clusters have diphosphines instead of phosphines for which the single P atoms are added in pairs by restricting the distance to a range of 5.0–5.5 Å based on the length of the carbon chain between the phosphine ends in the molecule (see Fig. 2). The spatial fitting of the organic groups (e.g., triphenyl) were done by one large spherical probe in parallel direction to the nearest neighbor Ag–P bond.

The results for the Au–P interface are shown in the Supplementary Table 6 and the P-atom positions in the best structures in Fig. 5. For all the clusters the success of prediction is perfect: 100% out of all 3072 model structures were correctly built. The prediction of phosphines is easier compared to thiolates due to the diminished number of possible local conformations. In general, phosphines tend to bind into a tetrahedral arrangement with respect to the nearest neighbor Ag-atom and the three nearest S-atoms as shown in Fig. 2. To describe this kind

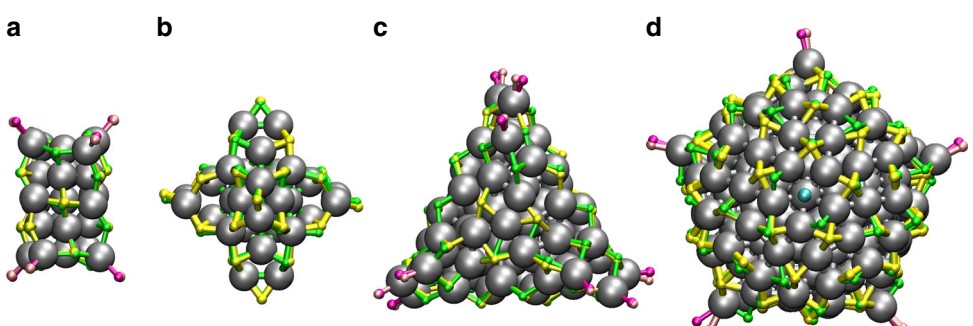

**Fig. 4** Determination of the true structure of Au-clusters using CSE ranking. Ratio of the correctly predicted S-atoms as a function of CSE for ligand-protected Au-clusters: **a** $Au_{34}(SR)_{22}$, **b** $Au_{36}(SR)_{24}$, **c** $Au_{38}(SR)_{24}$, **d** $Au_{44}(SR)_{26}$, **e** $Au_{52}(SR)_{32}$, **f** $Au_{92}(SR)_{44}$, **g** $Au_{102}(SR)_{44}$, **h** $Au_{146}(SR)_{57}$, and **i** $Au_{279}(SR)_{84}$. CSE includes contributions from error of local environments of Au and S atoms as well as from the number of predicted S atoms in units and in total

**Fig. 5** Validation of structure prediction for Ag-clusters. Comparison of the predicted and true ligand-metal interface structures of protected Ag-clusters: **a** $Ag_{23}(SR)_{18}(PPh_3)_8$, **b** $Ag_{44}(SR)_{30}$, **c** $Ag_{78}(SR)_{42}(DPPP)_6$, and **d** $Ag_{211}Cl(SR)_{71}(PPh_3)_6$. For predicted clusters Ag–S bonds are drawn in green and true Ag–S bonds in yellow whereas Ag–P bonds are drawn in magenta for predicted structure and in pink for the true structure. In panel (**d**) chlorine atom is drawn with light blue color

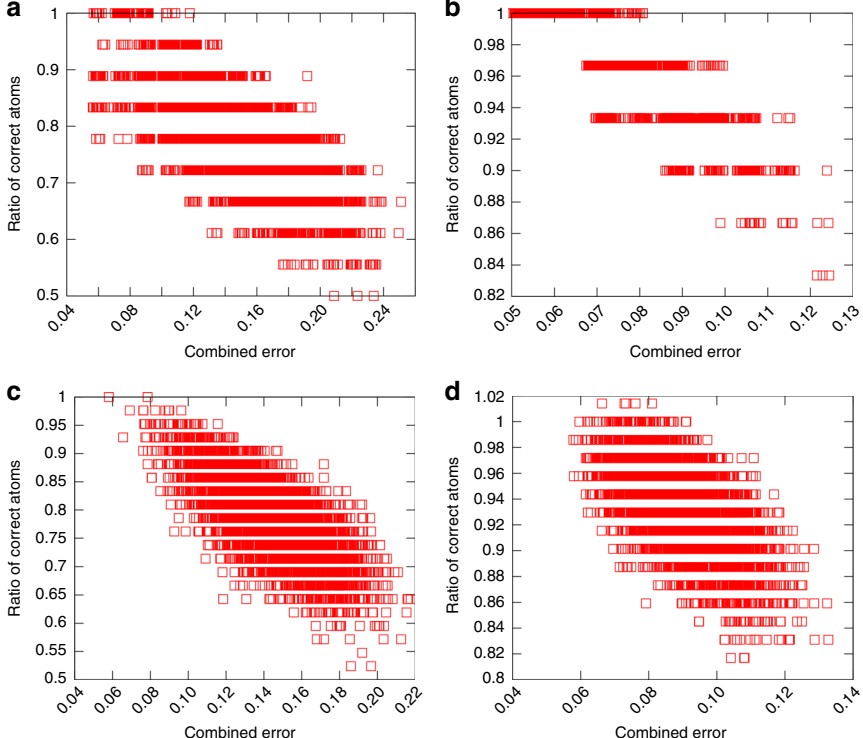

**Fig. 6** Determination of the true structure of Ag-clusters using CSE ranking. Ratio of correctly predicted S-atoms as a function of CSE for ligand-protected Ag-clusters: **a** $Ag_{23}(SR)_{18}(PPh_3)_8$, **b** $Ag_{44}(SR)_{30}$, **c** $Ag_{78}(SR)_{42}(DPPP)_6$, and **d** $Ag_{211}Cl(SR)_{71}(PPh_3)_6$. CSE includes contributions from error of local environments of Ag and S atoms as well as from the number of predicted S atoms in units and in total

tetrahedral arrangement, at least four nearest neighbors were needed for the local environment description.

**Correlation of CSE to DFT total energy.** It is interesting to correlate the CSE values to the DFT total energies for a group of generated model structures. We investigated this in the case of those $Au_{36}(SR)_{24}$ and $Ag_{44}(SR)_{30}$ model structures that have the molecular composition of the true cluster but not necessarily the true structure. The correlation of the CSE to the total energy is shown in Fig. 7a, b for gold and silver, respectively. The result shows a clear trend: the total energy decreases when the CSE decreases. This confirms that the minimum energy structures can be found most probably among the model structures that have low CSE values. The larger fluctuations at low and high energies seen in Fig. 7 are related to the decreased number of model clusters in sampling on both extremes. This comparison confirms the validity of CSE in predicting the true structure.

**Size effects of the ligands in Au₄₄ clusters.** The ligand size (steric volume) is one of the properties affecting the possible S- or P-atom positions on the metal surface. There are several examples of protected Au-clusters for which isomeric structures with different geometries are found with ligands of different bulkyness. Even cluster transformations driven by a ligand exchange from non-bulky to bulky ligand have been reported. Our algorithm provides a possibility to qualitatively study and understand the triggering conditions for these experimental findings. As an example, two different experimentally known thiolate-protected Au₄₄ clusters have been reported: one with 26 and the other with 28 thiolates from which the first is made with tertbutylbenzenethiol (TBBT) and the second with dimethylbenzenethiol (DMBT)[17,23]. These two clusters are completely different by the metal core symmetry and metal-ligand interface structures. Here we tested our algorithm whether it can predict that the true

structure and composition of the $Au_{44}(SR)_{28}$ is achieved only for TBBT ligand and not for the larger DMBT. The parameters used in the prediction and the results are summarized in Supplementary Tables 7 and 8. By varying the radius of the spherical probe from 2.5 Å (for modeling TBBT) to 3.0 Å (DMBT), the maximum number of thiolates at the interface drops from 28 to 26 (Fig. 8). The largest ligand that may occupy full 28 ligand sites on the surface corresponds to a 2.8 Å sized spherical probe. Remarkably, this in a perfect match with the true experimentally observed composition of DMBT protected Au-cluster. This can be understood by a competing effects from the interactions of the metal core and the metal-ligand interface and the spatial fitting of organic ligands, both affecting the overall structure. Since the metal core size is almost the same for both clusters and the metal atoms tend to maximize the packing of the core, there is roughly the same amount of free space for the ligands on the surface. This free space gets filled with 26 DMBT ligands if proper parameters for its steric volume are used.

**Discussion**

In this work, we have introduced a general method that can be used to predict metal-ligand interface structures of ligand-protected metal nanoparticles. The method uses the information from the local bonding environments of known reference structures (in our case the reference structures comprise reported crystal structures of similar thiolate- and phosphine-protected gold and silver nanoclusters) and can be easily generalized for the structural prediction of any nanostructure, in case enough reference information is available. The main variables, nearest neighbour bonds and bond angles between the interface atoms, are generally valid to be used for any atom type in any nanostructure. The steric parameters used in this paper can straightforwardly be generated to any atom types and molecular groups at the metal-ligand interface. Our algorithm is written in a

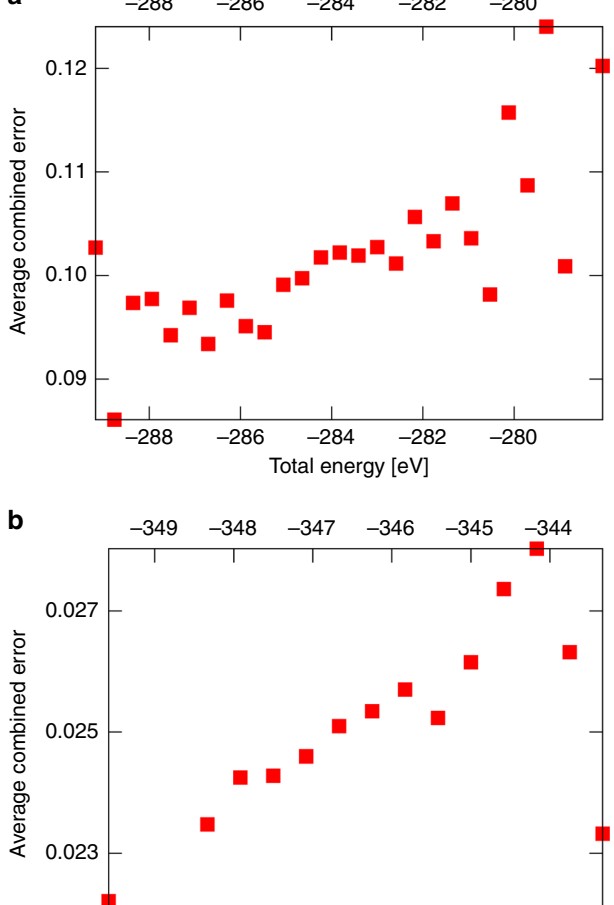

**Fig. 7** CSE correlates with DFT total energy. CSE as a function of the DFT total energy of the model structures of **a** $Au_{36}(SR)_{24}$ and **b** $Ag_{44}(SR)_{30}$ clusters. In both cases, 700 model structures that had the correct number of ligands were chosen for the analysis. CSE was calculated as an average over 0.4 eV energy range. For the calculation of the total energy, the SR-group was simplified with the SH-group as described in "Methods"

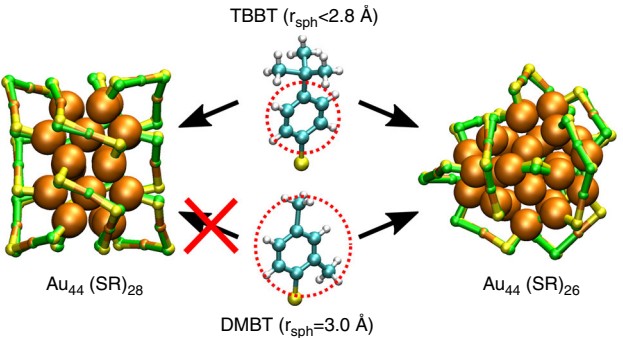

TBBT ($r_{sph} < 2.8$ Å)

$Au_{44}(SR)_{28}$          $Au_{44}(SR)_{26}$

DMBT ($r_{sph} = 3.0$ Å)

**Fig. 8** Size of the ligands affects the structure. Predicted model structures of $Au_{44}(SR)_{26}$ and $Au_{44}(SR)_{28}$ clusters for different ligands. Due to spatial fitting of the organic part of the ligand $Au_{44}(SR)_{28}$ is found only for ligand sizes <2.8 Å as mimicked by the spherical probe (see Supplementary Table 6). The maximum number of ligands with 3.0 Å probe on $Au_{44}(SR)_{28}$ surface is 26

modular way in order to maximize the flexibility and transferability to other metal-ligand systems, such as gold-alkynyls etc. An interesting test case would be provided by predictions of gold-thiolate interfacial structures in planar, self-assembling thiol monolayers on Au(111), which has been under intense investigations since 1980's[28,29].

We validated the method by predicting the Au–S interface structures of 10 known thiolate-protected gold nanoclusters and the Ag–S and Ag–P interface structures of four known ligand-protected silver clusters. Furthermore, we introduced a CSE parameter to measure the goodness of generated model clusters showing a clear correlation of low CSE value to low DFT total energy. The definition of CSE enables additional terms or fitness functions including both local and global descriptors, which could relate to structure-dependent properties such as X-ray powder diffraction patterns.

In all studied cases, the best-ranked structures essentially reproduced the bonding configuration of metal-ligand interfaces found in the crystal structure, with minor RMSD values in the predicted interface atom positions with respect to the crystal structure. We successfully predicted also the metal-ligand interfaces consisting of different kind of ligands (thiolates and phosphines), for which the prediction was made in steps by first predicting the most stable configurations of thiolates that are in majority and then predicting the positions of the phosphines that were in minority. In general, we expect that by applying structure optimization methods, a majority of the best model structures would relax to the known experimental structure. For the largest clusters studied in this work we also expect that a number of predicted best-ranked interface structures would lead to locally stable low-energy isomers lying energetically close to the known crystal structure. In this sense, our method should be useful in producing a number of potential structural isomers in a systematic and computationally effective way. These isomeric structures can then be examined with more robust energy-optimization methods such as DFT or DFT-based tight-binding methods. Since the spatial constraints from the ligand layer (i.e., the steric volume of the ligand molecule) are also parametrically included in the algorithm, we showed that our method can provide qualitative understanding on how the bulkiness of the ligand affects structures, interface conformations and compositions of the protected clusters.

Our experience on this method implies three critical points of concern to be adjusted to the system under investigation, to guarantee the success of the structure-prediction algorithm. First, a large enough sampling of the plausible local structures is needed in the set of reference structures. Second, the interval between grid points (Supplementary Fig. 1) has to be chosen fine enough. In this work, we used a value of 0.2 Å. Third, too loose criteria for describing the local environments of atoms may lead to improper bonding configurations that deviate from the true metal-ligand chemistry. We found reasonable to allow up to about 10% error in S–Au bond length and in RS–Au–SR angles depending on the number of nearest neighbors considered in the description. All the parameters we used were based on statistical analysis of experimental structural data of a similar class of clusters in question.

Unguided stochastic process starts to dictate the generation of the model structures for larger systems so that ever larger number of model structures have to be generated in order to have a complete representation of all relevant overall conformations. A further advantage of our method is that the process can be made guided by weighting the good choices made in model structure generation. In this study we introduced one possible approach for optimizing the global structural search. The main idea is to repeat the predictions by taking for the next run the possible interface atom positions from the set of best model structures of the

previous run. Other possibilities would be to weight the interface atom positions on-the-fly based on the success of the model structure generation.

Our future work will be directed to developing more numerically effective methods for evaluating the candidate structures, which eventually could reduce the need to use a large number of heavy total energy evaluations at the DFT level. Our goal is also to extend the structural prediction of metal-ligand interfaces into the metal atoms and complete cluster structures, bypassing then the need to parametrize classical force fields for complex hybrid nanomaterials. We hope that the method described in this work can open new avenues for effective structural predictions of nanoparticles and more generally nanomaterials where the atomic-scale information of the metal-ligand interface is crucial to understand growth mechanisms, stability, dynamics and ensuing physico-chemical properties. As such, our work is complementary to recent efforts to develop understanding of gold nanoparticle synthesis via deep learning[30].

## Methods

**General.** The logic of our method is described in the main text (steps 0–4 in the beginning of "Results" section) and the corresponding algorithms (algorithms 1–4) are given in the Supplementary Information. The physico-chemical reasons for the selected parameters used in the prediction algorithm are discussed in the Supplementary Note 2. The approach to use a greedy enlargement during the search resembles the classical graph traversal algorithms[31]. Such approaches form part of the search-based artificial intelligence as suggested by Nilsson[32].

**DFT total energy calculations for CSE correlations.** The DFT calculations were run using the GPAW code-package[33] with the grid spacing of 0.2Å and Perdew-Burke-Ernzerhof (PBE) xc-functional[34]. The total energies were calculated without structure relaxation from a set of 700 generated model structures of both $Au_{36}(SR)_{24}$ and $Ag_{44}(SR)_{30}^{4-}$ clusters by adding the ligands as simplified SH-groups in the most natural bonding direction using the optimal bond distance. Total energies were used for studying the correlation to the CSE, which was averaged in increments of 0.4 eV of total energy.

## Data availability

The algorithm published in this work is fully documented in the main text and in the Supplementary Information. All the reference structural data was taken from previously published work and referenced accordingly.

## Code availability

The software and examples of full datasets/runs for the examined cases that validated our method are available by request to H.H.

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

## Acknowledgements

This work was supported by Academy of Finland through the AIPSE research program, grants 315549 (H.H.), 315550 (T.K.) and 311877 (T.K.), and through H.H.'s Academy Professorship. The computations were done at the Nanoscience Center (NSC) of University of Jyväskylä, at the CSC supercomputing center in Finland and as part of a PRACE project in the Barcelona Supercomputing Center.

## Author contributions

S.M. and H.H. conceived the concept, S.M. built the structure prediction algorithm, validated the results and wrote the first draft of the manuscript. A.P., P.N., J.H. and T.K. contributed in the development of the algorithm. H.H. supervised the work. All authors commented on the manuscript draft that was finalized by H.H.

**Additional information**

**Competing interests:** The authors declare no competing interests.

**Peer Review Information:** *Nature Communications* thanks the anonymous reviewers for their contribution to the peer review of this work. Peer reviewer reports are available.

