## [Peer Review File · Nature Communications]

Reviewers' comments:

Reviewer #1 (Remarks to the Author):

The manuscript by Malola et al. describes a method to search for likely thiolate ligand arrangements on gold nanoparticles (NPs). The work is well presented and, as far as I am aware, this is the first example of an attempt to do this. The algorithm employed bases its primary search for sulphur positions on comparison with a set of reference ligated NP structures. Second, it proceeds with a stochastic search restricted by chemical heuristics (e.g. coordination, atomic distances). Finally, the ligated NP structures produced are ranked according to two empirical criteria. The method is applied to five ligated Au NPs and the method is shown to be quite successful in finding structures close to the experimental structure, when the latter is known. This approach to structure prediction depends on a number of specific choices, which although perhaps chemically reasonable, are not clearly shown to be applicable to other ligated NP systems, as suggested. As such, the generality of the approach is not established. In fact, to show only five cases of a successful application of the method to one type of system does not provide a particularly large set of confirming examples for general application to Au-thiolate NPs. This potential lack of generality is underlined by the fact that each example uses specifically tuned parameters (e.g. error margins for distances and angles, number and type of structures in reference set) which are reported without justification. The dependency of the success of the method on such particular choices should be provided.

Apart from the rather specific and empirical nature of the approach, perhaps its main limitation is the fact that it requires the positions of the Au atoms in the NP to be known prior to a search. In addition the method relies on a reference set of experimentally refined ligated-NP structures. In other words the searches only consider different ligand positions on known NP core structures biased by experimental knowledge of similar systems. Thus the method is limited in the scope of its predictive potential. Are there many cases where the positions of the metal atoms are well known in detail but where the X-ray positions of the ligands are not AND for which a number of experimental structures of ligated NPs are already known? If such scenarios are not common what is the main application of the method? The restriction of using a fixed known NP core also greatly limits the configurational search space and thus wonders if a simple force field based global optimisation approach of for finding low energy ligand binding positions on a fixed NP structure would be more appropriate and direct. What is the advantage of the current method over such direct, less empirical, approaches? The authors claim that their empirical ranking of obtained structures correlates well with that of energetic stability but that confirmation of this is outside the scope of the current work. I strongly feel that to help provide evidence that the ranking employed is reliable this correlation should be reported in the present work.

Predictive structure searches have been widely employed in many fields of chemistry and nanoscience. Although, the application of a bottom-up search algorithm for ligated NPs appears quite novel, the reported method is limited to known NP structures and, for each search, relies on a number empirical principles and tailored parameters. Although perhaps useful for the reported system, it is not convincing that the approach is as general or predictive as suggested in the title of the manuscript. Considering the above, I feel that the current manuscript would be better suited for publication in a more specialised journal.

Reviewer #2 (Remarks to the Author):

This report describes the use of computationally inexpensive algorithms for predicting the structure of gold monolayer protected clusters (MPCs) based on a chemical basis set of known structure solutions. The authors were able to successfully generate the ligand structure of five clusters capped with different thiolate ligands by starting with the known gold core structure; this included clusters ranging from 36 to 279 gold atoms. The structural features of double and/or

triple bridged Au(I)thiolate (staples and hemirings) were observed in all of the MPCs with single bridge ligands only being observed occasionally in some clusters. These results are expected for this class of MPCs and demonstrate the validity of their algorithms and choice of input parameters.

This work is of great significance as a means for general structure prediction, although it has only been demonstrated on Au-S based MPCs here. The adaptability of this method to include partial structure information from experimental data will be especially helpful in cases where total structure determination by XRD is challenging. Overall, this work is of the appropriate caliber and scope for publication in this journal.

Questions/Points of Clarification:

The algorithm success rate for Au₃₈(SR)₂₄ was dramatically higher than for any other cluster, even compared to Au₃₆(SR)₂₄, which used the same basis set (albeit a different ligand). Can the authors comment on this?

Page 7, Au₃₆(SR)₂₄ section: the ligand is listed as tert-butyl benzene (TBBT) instead of tert-butyl benzene thiol. I'm assuming this is a typo.

Page 10, Discussion: paragraph 2, Au-S "bonding configuration" – please define this, or is it supposed to be bonding configuration?

Page 14, 1. Local Environments: "extending the NN distances to 3-4 takes into account" I am unclear if this means 3-4 angstroms or if kNN = 3-4? Can you clarify this part?

Reviewer #3 (Remarks to the Author):

The authors presented a computational method to predict the ligand-metal layers of thiolate-protected gold nanoclusters. A few successful examples were explored by the algorithm. It is always attractive to obtain the structure of a nanocluster without growing single crystals, however, I do not think the manuscript is suitable to publish on Nat. Comm., unless the authors can address the following concerns:

1. The manuscript focuses on the thiolate-protected nanoclusters, but in fact, other types of ligands, such as alkyne, phosphine, amine, pyridine, or a mixture of them, have also been used to protect the metal core. Since other types of ligands have very different coordination environment than thiolate (different staples, or no staples at all), I wonder whether the method presented here can be applied to other systems easily as the authors claimed.
2. In order to predict the structure of metal-ligand layers successfully, it seems that all gold atoms, including the ones that forms the staples, have to be fixed before adding thiolate ligands. However, the identification of surface gold atoms or even core gold atoms has already troubled the experimental research a lot. I think the significance is not as great as it looks, because people will still have to spend enormous amount of time to obtain the location of metal atoms.
3. There are several examples of polymorphism thiolate-protected gold nanoclusters, such as Au₃₀, Au₃₈, Au₄₄, Au₅₂. I think they might be better candidates to be explored theoretically, because they show that different types of thiolate ligands can cause different metal-ligand interface, as well as different core structures.

RESPONSE TO REFEREES

We thank all the referees for their constructive criticism and insightful comments. Below, we repeat the comments and give a point-by-point response (*italics*).

Referee 1

1. The manuscript by Malola et al. describes a method to search for likely thiolate ligand arrangements on gold nanoparticles (NPs). The work is well presented and, as far as I am aware, this is the first example of an attempt to do this. The algorithm employed bases its primary search for sulphur positions on comparison with a set of reference ligated NP structures. Second, it proceeds with a stochastic search restricted by chemical heuristics (e.g. coordination, atomic distances). Finally, the ligated NP structures produced are ranked according to two empirical criteria. The method is applied to five ligated Au NPs and the method is shown to be quite successful in finding structures close to the experimental structure, when the latter is known. This approach to structure prediction depends on a number of specific choices, which although perhaps chemically reasonable, are not clearly shown to be applicable to other ligated NP systems, as suggested. As such, the generality of the approach is not established. In fact, to show only five cases of a successful application of the method to one type of system does not provide a particularly large set of confirming examples for general application to Au-thiolate NPs. This potential lack of generality is underlined by the fact that each example uses specifically tuned parameters (e.g. error margins for distances and angles, number and type of structures in reference set) which are reported without justification. The dependency of the success of the method on such particular choices should be provided.

Response: We thank the referee for comments on the presentation and novelty of our work. In the revision, we have significantly increased the number of tested systems, from five to ten Au-SR interfaces, as well as generalized the discussion to several Ag-SR and mixed Ag-SR-PR₃ interfaces. Furthermore, we show a successful case where the algorithm predicts correctly steric effects of thiolates on the metal-ligand interface of Au₄₄(SR)_{26/28} clusters. Discussion on the criteria of selecting the geometrical parameters to define local chemical environments of interface atoms guiding the structural search has been extended (Supplementary Note 3), and we show that our selected parameters are based on a statistical analysis of such interfaces from extensive experimental data.

2. Apart from the rather specific and empirical nature of the approach, perhaps its main limitation is the fact that it requires the positions of the Au atoms in the NP to be known prior to a search. In addition the method relies on a reference set of experimentally refined ligated-NP structures. In other words the searches only consider different ligand positions on known NP core structures biased by experimental knowledge of similar systems. Thus the method is limited in the scope of its predictive potential.

Response: We agree that the predictive power of the method concerning “novel” metal-ligand interfaces structures is ultimately defined by the extent by which the already known experimental structures have been able to catch the essential local chemistry around the metal atoms and protecting ligands. We dare to argue that this most probably is the case at least in the best-studied gold-thiolate systems, where around 100 crystallographically determined cluster structures exist in the literature.

Postal address:

Nanoscience Center, P.O. Box 35
University of Jyväskylä, FI-40014 Jyväskylä, Finland

Tel: +358-400-247973

<http://users.jyu.fi/~hahakkin>
hannu.j.hakkinen@jyu.fi

Chemists use frequently analogues in predicting chemical binding. It is already known from a much less abundant database of gold-alkynyl clusters, that the binding modes of alkynyls to gold seem to be very similar to thiolates. Thus we expect that the method would work equally well for such systems. Recent work has also shown analogues in metal-ligand binding geometries between gold-phosphine and gold-carbene clusters, where the interface structure is much simpler compared to gold-thiolates and gold-alkynyls. Silver-thiolate interfaces are more complex than gold-thiolates due to the larger range of possible metal-ligand coordination geometries. It is clear that successful use of our algorithm requires some chemical knowledge of the system under study, but this requirement is trivially true for a successful use of any predictive computational method.

3. Are there many cases where the positions of the metal atoms are well known in detail but where the X-ray positions of the ligands are not AND for which a number of experimental structures of ligated NPs are already known? If such scenarios are not common what is the main application of the method? The restriction of using a fixed known NP core also greatly limits the configurational search space and thus wonders if a simple force field based global optimisation approach of for finding low energy ligand binding positions on a fixed NP structure would be more appropriate and direct. What is the advantage of the current method over such direct, less empirical, approaches?

Response: *Despite the great recent successes of atom-precise experimental structural characterizations of ligand-stabilized metal nanoclusters by single-crystal X-ray crystallography, there are still many cases (and more expected) where the atom-precise total structure determination, including the organic ligand layer, may not succeed. In fact, the initial motivation for us to start developing the method described in this work came from the collaboration with the group of R.D. Kornberg (Science 345, 909 (2014); ACS Nano 11, 11866 (2017); ACS Nano 11, 11872 (2017)). Kornberg's group had for years tried to successfully determine total structures by X-ray diffraction of a class of gold clusters stabilized by certain thiolates, but without success. They resolved to use cryo-EM 3D tomography that yielded the 3D coordinates of the gold core but no information about the structure of the ligand layers. Complementary mass spectrometry data yielded the number of ligands, i.e., the total chemical composition of the clusters was then known. This information was the starting point for our computational work. In situations like this, our method can work in a very important role in providing a number of good candidates for initial structures to be then refined with force-field, DFT, etc.. methods depending on the case. In the revised MS we have introduced ranking criteria and showed a strong correlation between the best structures from the ranking and the lowest DFT-computed total energies (see our next point), to validate this strategy. The computational burden of our method is several orders of magnitude less than DFT-based optimization methods, facilitating a significantly faster and more diverse search of initial structure candidates.*

On the experimental front, it can be expected that the use of various high-resolution EM techniques, possibly coupled with tomography, is going to increase also in the field of structural studies of various metal-ligand nanostructures. EM will always have fundamental challenges to detect light organic molecules bound to much more electron-rich metals. Thus we expect that methods like ours will in fact be useful in

Postal address:

Nanoscience Center, P.O. Box 35
University of Jyväskylä, FI-40014 Jyväskylä, Finland

Tel: +358-400-247973

<http://users.jyu.fi/~hahakkin>
hannu.j.hakkinen@jyu.fi

connection with high-resolution experiments that can provide only partial information about the atom-precise structure of a nanosystem.

4. The authors claim that their empirical ranking of obtained structures correlates well with that of energetic stability but that confirmation of this is outside the scope of the current work. I strongly feel that to help provide evidence that the ranking employed is reliable this correlation should be reported in the present work.

Response: We have now introduced a new fitness parameter to rank the model structures, a so-called Combined Structural Error (CSE). Figures 4 and 6 show a strong correlation of the CSE-ranked best model structures to the true crystallographic structure of several gold and silver clusters, respectively, and Figure 7 shows that the best CSE-ranked model structures of $Au_{36}(SR)_{24}$ also have the lowest calculated DFT total energies. Please note that our definition of the CSE is straightforward to generalise to include other desired criteria, such as "closeness" of a simulated powder XRD function of a model structure to experimentally measured XRD data, etc.

5. Predictive structure searches have been widely employed in many fields of chemistry and nanoscience. Although, the application of a bottom-up search algorithm for ligated NPs appears quite novel, the reported method is limited to known NP structures and, for each search, relies on a number empirical principles and tailored parameters. Although perhaps useful for the reported system, it is not convincing that the approach is as general or predictive as suggested in the title of the manuscript. Considering the above, I feel that the current manuscript would be better suited for publication in a more specialised journal.

Response: As already discussed above, we have extended the analysis in the revised MS to include significantly more and different systems as compared to the original version of the MS. It should be noted that, in principle, our method is not limited to clusters or nanoparticles having metal-ligand interfaces. It can be used as well for planar metal-molecule interfaces, such molecular SAMs, or complicated 3D systems such as molecule-capped metal nanowires, metal-organic networks, etc.

Referee 2

1. This work is of great significance as a means for general structure prediction, although it has only been demonstrated on Au-S based MPCs here. The adaptability of this method to include partial structure information from experimental data will be especially helpful in cases where total structure determination by XRD is challenging. Overall, this work is of the appropriate caliber and scope for publication in this journal.

Response: We thank the referee for his/her positive evaluation of the significance of our work.

2. The algorithm success rate for $Au_{38}(SR)_{24}$ was dramatically higher than for any other cluster, even compared to $Au_{36}(SR)_{24}$, which used the same basis set (albeit a different ligand). Can the authors comment on this?

Response: *This is indeed true and intriguing. $Au_{38}(SR)_{24}$ seems to have a “simple” and “well-coordinated” protective ligand layer. It is very intriguing to realize that it is experimentally known that $Au_{38}(SR)_{24}$ is chemically a very stable cluster. In fact, the well-known $Au_{25}(SR)_{18}$ cluster spontaneously transforms to $Au_{38}(SR)_{24}$ if left in solution. However, we do not have space to discuss this further in the MS.*

3. Page 7, $Au_{36}(SR)_{24}$ section: the ligand is listed as tert-butyl benzene (TBBT) instead of tert-butyl benzene thiol. I'm assuming this is a typo.
4. Page 10, Discussion: paragraph 2, Au-S “bonding configuration” – please define this, or is it supposed to be bonding configuration?
5. Page 14, 1. Local Environments: “extending the NN distances to 3-4 takes into account” I am unclear if this means 3-4 angstroms or if $kNN = 3-4$? Can you clarify this part?

Response: Due to the extensive rewriting of the Results section, these comments are not valid anymore.

Referee 3

1. The authors presented a computational method to predict the ligand-metal layers of thiolate-protected gold nanoclusters. A few successful examples were explored by the algorithm. It is always attractive to obtain the structure of a nanocluster without growing single crystals, however, I do not think the manuscript is suitable to publish on Nat. Comm., unless the authors can address the following concerns:

Response: *We have now significantly revised the MS including many new test cases that show the applicability and performance of our method.*

2. The manuscript focuses on the thiolate-protected nanoclusters, but in fact, other types of ligands, such as alkyne, phosphine, amine, pyridine, or a mixture of them, have also been used to protect the metal core. Since other types of ligands have very different coordination environment than thiolate (different staples, or no staples at all), I wonder whether the method presented here can be applied to other systems easily as the authors claimed.

Response: *We have now included two other metal-ligand interfaces (Ag-SR and mixed Ag-SR- PR_3) as well as increased the number of examined Au-SR systems from five to ten. We show that the algorithm works well for all these systems. As noted in our response to referee 1 (see point 2), we expect the method to be straightforward to use for many other metal-ligand interfaces as well. Obviously this is the very first MS describing the method and showing the first test case studies and we cannot cover all possible metal-ligand interface systems here.*

3. In order to predict the structure of metal-ligand layers successfully, it seems that all gold atoms, including the ones that forms the staples, have to be fixed before adding thiolate ligands. However, the identification of surface gold atoms or even core gold atoms has already troubled the experimental research a lot. I think the significance is

Postal address:

Nanoscience Center, P.O. Box 35
University of Jyväskylä, FI-40014 Jyväskylä, Finland

Tel: +358-400-247973

<http://users.jyu.fi/~hahakkin>
hannu.j.hakkinen@jyu.fi

not as great as it looks, because people will still have to spend enormous amount of time to obtain the location of metal atoms.

Response: *Please see our response to Referee 1, comment 3.*

4. There are several examples of polymorphism thiolate-protected gold nanoclusters, such as Au₃₀, Au₃₈, Au₄₄, Au₅₂. I think they might be better candidates to be explored theoretically, because they show that different types of thiolate ligands can cause different metal-ligand interface, as well as different core structures.

Response: *We thank the referee for this very good suggestion. We now discuss a case where the algorithm predicts correctly steric effects of thiolates on the metal-ligand interface of Au₄₄(SR)_{26/28} clusters. However length limitations preclude us from including further examples of this class.*

REVIEWERS' COMMENTS:

Reviewer #1 (Remarks to the Author):

The authors have made significant improvements to the manuscript which helps to strengthen their claims. In particular, the addition of new Au and Ag nanoparticle systems helps to confirm the applicability of their approach to a wider range of systems. The more precisely defined CSE ranking, and showing (at least in the case of the Au₃₆(SR)₂₄ system) that it correlates quite well with DFT-evaluated energetic stability, also supports the efficacy of their method. Here, I recommend that the authors also include a similar plot for the case of the Ag₂₃(SR)₁₈(PPh₃)₈ system to further confirm the generality of the CSE ranking being a good measure/descriptor of energetic stability. In their response the authors note that their method is significantly more computationally efficient than DFT. However, it is not clear that it would have a significant computational advantage over a classical forcefield based global optimisation approach. This said, the present method avoids the complication of parameterising suitably accurate forcefields for each new system - which the authors may consider mentioning.

Although I am still not fully convinced as to the extent of the generality of the method proposed (and thus still suggest to remove the word "general" from the manuscript title) the authors have responded in full to most of my comments and the manuscript is now much more complete and convincing. In my opinion the revised manuscript is now suitable for publication in Nature Communications.

Reviewer #3 (Remarks to the Author):

I would like to thank the authors for the tremendous efforts that have been done during the revision. The authors have provided convincing discussions to show the research is suitable for publication. The manuscript should be accepted in its current shape.

RESPONSE TO REFEREES

Below, we repeat the comments by Referees 1 and 3, and give our response (*italics*).

Referee 1

The authors have made significant improvements to the manuscript which helps to strengthen their claims. In particular, the addition of new Au and Ag nanoparticle systems helps to confirm the applicability of their approach to a wider range of systems. The more precisely defined CSE ranking, and showing (at least in the case of the Au₃₆(SR)₂₄ system) that it correlates quite well with DFT-evaluated energetic stability, also supports the efficacy of their method. Here, I recommend that the authors also include a similar plot for the case of the Ag₂₃(SR)₁₈(PPh₃)₈ system to further confirm the generality of the CSE ranking being a good measure/descriptor of energetic stability.

We have now done the CSE vs. total energy – analysis also for the Ag₄₄(SR)₃₀ system (to make it easier to compare to the Au₃₆(SR)₂₄ system). The results are shown in the new Fig. 7b and mentioned in the main text.

In their response the authors note that their method is significantly more computationally efficient than DFT. However, it is not clear that it would have a significant computational advantage over a classical forcefield based global optimisation approach. This said, the present method avoids the complication of parameterising suitably accurate forcefields for each new system - which the authors may consider mentioning.

We thank the Referee for this suggestion and have noted this in the Discussion section.

Although I am still not fully convinced as to the extent of the generality of the method proposed (and thus still suggest to remove the word "general" from the manuscript title) the authors have responded in full to most of my comments and the manuscript is now much more complete and convincing. In my opinion the revised manuscript is now suitable for publication in Nature Communications.

We have changed the title as suggested. Finally, we thank the Referee for his/her valuable help to improve our paper.

Referee 3

I would like to thank the authors for the tremendous efforts that have been done during the revision. The authors have provided convincing discussions to show the research is suitable for publication. The manuscript should be accepted in its current shape.

We thank the Referee for his/her valuable help to improve our paper.

Postal address:

Nanoscience Center, P.O. Box 35
University of Jyväskylä, FI-40014 Jyväskylä, Finland

Tel: +358-400-247973

<http://users.jyu.fi/~hahakkin>
hannu.j.hakkinen@jyu.fi